# Diet as a Modulator of Gut Microbiota May Reduce Alzheimer’s Disease Risk

**DOI:** 10.3390/nu17193053

**Published:** 2025-09-24

**Authors:** Agnieszka Małgorzata Ochocińska, Izabela Podstawka, Alina Kępka, Napoleon Waszkiewicz

**Affiliations:** 1Department of Clinical Biochemistry, The Children’s Memorial Health Institute, 04-730 Warsaw, Poland; 2Department of Psychiatry, Medical University of Bialystok, 15-089 Bialystok, Poland; napwas@wp.pl

**Keywords:** Alzheimer’s disease, diet, gut–brain axis, microbiome, neurodegeneration

## Abstract

The aging process, along with an inadequate diet and an inflammatory gut response resulting from dysbiosis, contributes to the pathogenesis of Alzheimer’s disease (AD). Modifying the composition of the gut microbiota through appropriate pre/probiotic-rich diets may act as a preventive option for AD. The variety of functions performed by the gut microbiota makes this ecosystem one of the most important systems in the human body. The Mediterranean diet (MedDiet), the Dietary Approaches to Stop Hypertension (DASH), the Mediterranean–DASH Intervention for Neurodegenerative Delay diet (MIND), and the modified ketogenic–Mediterranean diet (MKD) positively affect the intestinal microflora and may reduce the risk of dementia. A ketogenic diet has a neuroprotective effect and improves cognitive function but leads to a significant decrease in the abundance and diversity of bacterial species in favor of harmful bacteria. A Western-style diet (Western diet, WD) rich in processed products, red meat, simple sugars, and saturated fatty acids has a negative impact on gut microbiota function, increasing the risk of AD. Our review supports the hypothesis that factors like a proper diet and a healthy gut microbiota have a positive impact on the prevention of neurodegenerative diseases, including AD. A thorough understanding of the role the microbiota plays in the proper functioning of the nervous system can aid in the prevention of AD by developing new dietary strategies and dietary lifestyles.

## 1. Introduction

Alzheimer’s disease (AD) is a chronic, progressive, neurodegenerative disease of the brain that causes nerve cell loss. It is caused by inflammation of the nervous tissue and disturbances in the metabolism of β-amyloid precursor protein (βAPP). This leads to the extracellular accumulation of toxic insoluble deposits of the β-amyloid peptide (Aβ), mainly Aβ_40_ and, to a lesser extent, Aβ_42_/_43_, and the deposition of intracellular neurofibrillary tangles (NFTs) formed by hyperphosphorylated tau protein (τ) (a phosphoprotein that binds microtubules), as well as activation by Aβ of microglia, which perform immune and regulatory functions and participate in the formation of neural networks in the brain. Microglia immune cells stimulated over a long period of time trigger the chronic release of pro-inflammatory cytokines, which over the years lead to irreversible neurodegeneration, especially in the hippocampus. Soluble Aβ peptides interact with each other to form many forms of oligomers and aggregates that associate into insoluble fibrillar amyloid deposits, causing synaptic damage, neuronal dysfunction, neuroinflammation, and neurodegeneration, and they also participate in the hyperphosphorylation of the protein τ, which is deposited in the form of NFTs, leading to the disintegration of the neuronal skeleton, loss of dendritic spines, progressive synaptic dysfunction, and, ultimately, neuronal death [1,2]. Excessive amounts of other protein deposits, such as α-synuclein (ASN) and transactive response DNA-binding protein 43 kDa (TDP-43), play an important role in the pathogenesis of AD [3,4]. The main symptom of this disease is progressive dementia (a syndrome of cognitive and behavioral disorders that cause problems in performing everyday activities, such as eating, dressing, and toileting). It is estimated that AD accounts for 60–80% of all dementia cases [5].

The etiopathogenesis of AD is multifactorial. The following pathomechanisms are mentioned: biochemical (the deposition of amyloid deposits leads to a number of biochemical disorders in brain tissue), immunological (activation of non-specific immune mechanisms and age-related chronic inflammation in the body participate in neurodegenerative processes, resulting in the release of inflammatory factors, such as cytokines, chemokines, nitric oxide, and reactive oxygen and nitrogen species), and genetic. Genetic defects account for approximately 5% of all AD cases. Four genes are responsible for the inheritance of the disease: APOE (apolipoprotein E), PSEN1 (presenilin-1), PSEN2 (presenilin-2), and βAPP. A mutation in the PSEN1 gene is responsible for 20–70% of cases of early AD [6]. The late-onset familial form (>50 years of age) is associated with inheritance of the APOE gene. The ε4 isoform (encoded by APOE 4) increases the risk of developing the disease by up to several times. Other genes associated with late-onset Alzheimer’s disease are also known, e.g., ABCA7, CLU, CR1, PICALM, SORL1, TREM2, and PLD3 [7]. In addition to the factors mentioned above, mitochondrial dysfunction is also a mediator of Alzheimer’s disease pathogenesis. Mitochondria, cellular organelles, are responsible for producing the energy necessary for neurons to function. Mitochondrial dysfunction involves abnormal structure and dynamics, as well as disturbances in the functioning of the electron transport chain, the formation of excessive amounts of free radicals, and oxidative stress. Impaired levels of enzymes involved in the electron transport chain and the Krebs cycle can escalate pathological processes. Mitochondria play a key role in regulating calcium balance and apoptosis. Their dysfunction can lead to the development of inflammatory processes and progressive damage to nerve tissue. Age is an indisputable risk factor for AD. Other factors related to lifestyle and medical conditions include low educational attainment, obesity, depression, sleep disorders, hypertension, cerebrovascular disease (the most commonly reported risk factor), diabetes, dyslipidemia (elevated cholesterol levels may threaten the integrity of the blood–brain barrier), smoking, alcohol abuse, physical inactivity, and diet.

In recent years, particular attention has been paid to diets that could reduce the risk of AD. It is believed that a well-balanced diet containing anti-inflammatory, antioxidant, anti-amyloid, and neuroprotective ingredients can slow down the disease [8,9,10]. The Mediterranean diet (MedDiet), considered one of the healthiest diets, may have a preventive effect on the nervous system [11]. MedDiet is mainly based on plant-based foods, such as fruits and vegetables, but also whole grains, seeds, nuts, and olive oil. It is not recommended to consume butter, sugar (sweets), fatty meats, or meat products. Eggs, skimmed milk, and dairy products, as well as lean poultry, pork, and beef can be consumed in smaller quantities. Fish and seafood are recommended. Like the MedDiet, the Dietary Approaches to Stop Hypertension (DASH) diet is a source of antioxidants, B vitamins, polyphenols, PUFA (polyunsaturated fatty acids), MUFA (monounsaturated fatty acids), and DHA (docosahexaenoic acid), which can protect the brain and prevent Alzheimer’s disease [12].

The Mediterranean–DASH Intervention for Neurodegenerative Delay diet (MIND) is a combination of both the MedDiet and DASH diets, which is less restrictive and easier to follow than other diets. Regular use of the MIND diet leads to improved cognitive function [13]. Another way of eating is the ketogenic diet (KD), which is high in fat and low in carbohydrates, with an adequate amount of protein, leading to increased production of ketone bodies. It is used in the prevention and treatment of neurodegenerative diseases, including AD [14]. A diet that has a more beneficial effect on the human body is the Modified Mediterranean–ketogenic diet (MKD), which is a combination of the principles of the classic ketogenic diet with features of the Mediterranean diet, which has a preventive effect on many diseases, including neurodegenerative diseases, such as AD [15]. The Western diet (WD) is prevalent in Western Europe and the USA, characterized by the consumption of foods with low nutritional value and high calorie content, rich in fat, including saturated fatty acids, processed foods, simple sugars, and salt, and low in fibre. Following this diet may lead to an increased incidence of many diseases, including inflammation in the nervous system, which may contribute to the development of neurodegenerative diseases [16].

It has been proven that a proper diet based mainly on vegetables and fruits, legumes, products rich in polyphenols (berries, blackberries, blueberries, nuts, seeds, spices, olive oil, red grapes), fermented and pickled products (sauerkraut, kimchi, tempeh, natto), and foods rich in prebiotics (onions, garlic, chicory, leeks, bananas, asparagus, oats) as well as fermented milk products (kefir, yogurt, buttermilk) has a direct impact on the composition of the gut microbiota. In turn, the gut microbiota has a direct impact on brain health. In addition, a diet that is healthy for the gut and brain should include fish and seafood and low-fat dairy products. On the other hand, it should be low in red meat, poultry, high-fat dairy products, and processed foods (fried foods, salty snacks, sweets, sugar-sweetened beverages) (Table 1). More and more research shows a link between diet, the gut microbiome, and neurodegenerative diseases, including AD [17,18]. It has been confirmed that diet has a huge impact on the state of the gut microbiota, which is essential for the proper functioning of many of our systems, including the nervous system [19,20] (Figure 1). The gut microbiota regulates brain function through the microbiota–gut–brain axis, sending information bidirectionally through hormonal, neuronal, metabolic, and immunological mechanisms [5,21,22]. There is no doubt that many diseases, such as cancer, metabolic disorders, cardiovascular diseases, allergies, obesity, and even mental disorders (schizophrenia) and neurodegenerative diseases, are strongly associated with intestinal dysbiosis [2,23]. It is believed that the more diverse the diet, the greater the stability of the microbiome and the better brain health during aging. Recent studies confirm that the composition of the microbiome is unique to each individual, which means that in the future, individual dietary patterns aimed at modulating the gut microbiota may need to be tailored to each person’s personalized microbiome [24,25,26].

The literature emphasizes the importance of the gut microbiota, which is associated with factors determining susceptibility to AD. In patients with AD, it has been shown that the diversity, composition, and abundance of the gut microbiota differ significantly compared to healthy individuals. In AD, a reduced abundance of anti-inflammatory bacteria belonging to the phylum Firmicutes (changed to Bacillota in 2021, but most authors use the previous name) has been found, i.e., bacteria of the genus *Clostridium*, *Lachnospira*, and *Lachnoclostridium* and bacteria from the *Fusobacteriaceae* family, as well as *Actinobacteria*, e.g., *Bifidobacterium.* In AD, an increased number of pro-inflammatory bacteria, such as *Escherichia/Shigella* and *Klebsiella*, belonging to the phylum *Proteobacteria*, have been found. Amyloids secreted by bacteria, such as *Escherichia coli*, affect the abnormal folding of the alpha-synuclein protein, leading to an increased risk of AD [19,20,27,28,29]. The results regarding the abundance of *Bacteroidetes* in AD are inconclusive. There is no doubt that the abundance of *Prevotella* (family *Prevotellaceae*; phylum Bacteroidetes) increases with disease progression [30]. However, some studies have shown reduced levels of *Bacteroides* in AD patients compared to healthy individuals with normal cognitive function [31,32]. A decrease in the abundance of bacteria belonging to the phylum Firmicutes (Bacillota) in AD patients, including the genera *Ruminoclostridium 9*, *Ruminococcus*, *Clostridium*, *and Lachnospira*, leads to a decrease in SCFA production. There is a correlation between the abundance of bacteria, such as *Lachnospiraceae*, *Lachnoclostridium* spp. *Roseburia hominis*, and *Bilophila wadsworthia*, and the presence of phosphorylated protein τ. Significant accumulation of this protein was associated with a decrease in the abundance of the aforementioned bacteria [33]. In addition, other species belonging to Firmicutes (Bacillota), such as *Lactobacillus* producing GABA and acetylcholine, or *Clostridium sporogenes* producing indole propionic acid, which plays a neuroprotective role against cell damage and death caused by Aβ accumulation, are also reduced in patients with AD [27]. In contrast, an increase in the abundance of bacteria belonging to Firmicutes (Bacillota), including species like *Turicibacter* (family *Turicibacteraceae*) and *Clostridium leptum* spp. (family *Clostridiaceae*), and a decrease in the abundance of *Eubacterium ventriosum* spp. (family *Eubacteriaceae*), *Lachnospiraceae* and the genus *Marvinbryantia* belonging to this family, as well as *Monoglobus* (family *Oscillospiraceae*), *Runinococcus torques* spp. (family *Oscillospiraceae*), *Roseburia hominis* spp. (family *Lachnospiraceae*), and *Christensenellaceae R-7* spp. (family *Christensenellaceae*), are correlated with a greater presence of deposits in the cerebrospinal fluid [30,33]. In patients with Alzheimer’s disease, the relative abundance of butyrate-producing species, such as *Eubacterium rectale*, *Eubacterium eligens*, and *Eubacterium halli*, is reduced compared to healthy control subjects [27]. Furthermore, the gut microbiome of individuals with AD is characterized by a greater abundance of other amyloid fiber-producing bacterial species, namely *Bacillus subtilis*, *Salmonella enterica*, *Staphylococcus aureus*, and *Mycobacterium tuberculosis* [19,34].

The methods of this review article were based on the use of the PubMed, Cochrane, EBSCO, EMBASE, and Scopus databases to search for all related published studies. The selection was based on the keywords “Alzheimer’s disease”, “diet”, “gut-brain axis”, “microbiome”, and “neurodegeneration”.

## 2. The Gut Microbiota and the Central Nervous System

The multitude of functions performed by the gut microbiota makes this ecosystem one of the most important systems in the human body [19,35]. Communication between the gut and the brain is bidirectional via the gut–brain axis. The direct route of communication is the vagus nerve (made up of afferent and efferent fibers), which collects information from the visceral organs and transmits it to the brain and then provides feedback to these organs from the brain. An inadequate diet and other factors affecting afferent fibers disrupt the normal communication of the brain–gut–microbiota axis [5,36]. The indirect pathway of communication between the gut microbiota and the central nervous system (CNS) occurs, among other things, via the autonomic nervous system and endocrine, metabolic, and immune mechanisms [21]. The gut microbiota is a very important link in gut–brain communication. Bacteria form a complex ecosystem in the digestive system, accounting for 94–98% of all isolated microorganisms. The most numerous groups of bacteria are two types, Firmicutes (Bacillota) (64%) and Bacteroidetes (23%), with a small percentage of Proteobacteria (8%) and Actinobacteria (3%). These bacteria are present throughout the digestive tract, but the largest number colonize the large intestine [37]. Intestinal bacteria influence the CNS, among other things, through their ability to synthesize certain vitamins (K, B vitamins, e.g., thiamine, biotin, folic acid) and stimulate the immune system of the digestive tract, which plays an important role in local and general immunity of the body (they modulate the concentration of, among others, pro-inflammatory cytokines, e.g., TNF-alpha, IFN-gamma, IL-6, and anti-inflammatory cytokines, e.g., IL-10) [38]. They also participate in the production of a wide range of neurotransmitters and other compounds identical to those found in the brain, e.g., serotonin (5-hydroxytryptamine, 5-HT), acetylcholine (Ach), melatonin, γ-aminobutyric acid (GABA), catecholamines (adrenaline, noradrenaline, dopamine), histamine, corticotropin-releasing hormone (CRH), brain-derived neurotrophic factor (BDNF), synaptophysin, and postsynaptic density protein 95 (PSD-95) [38,39]. For example, certain species of *Lactobacillus* and *Bifidobacterium* synthesize GABA, *Escherichia*, *Bacillus*, and *Saccharomyces* spp. produce noradrenaline, *Candida*, *Streptococcus*, *Escherichia*, and *Enterococcus* spp. produce 5-HT, *Bacillus* produces dopamine, and *Lactobacillus* secretes ACh [21,40]. Another example is intestinal bacteria of the genus *Bacteroides*, such as *Bacteroides ovatus*, which are responsible for the production of 3-hydroxybenzoic acid (3-HBA) and 3-(3′-hydroxyphenyl) propionic acid (3-HPPA) by metabolizing catechins and epicatechins found in cocoa, green and black tea, red wine, fruits (apples, apricots, pears, plums, peaches, raspberries, blackberries, grapes, cherries), and vegetables (onions, broad beans, beans, rhubarb, capers, sorrel, dill). The metabolites listed above (3-HBA and 3-HPPA) have neuroprotective effects; crossing the blood–brain barrier, they can inhibit the formation of Aβ plaques and inhibit the formation of alpha-synuclein fibrils (alphaSN). In its physiological state, this protein is unfolded, but in its aggregated form, it takes on a structure known as a beta-hairpin, forming insoluble oligomer deposits that are involved in the process of cell death in neurodegenerative diseases. alphaSN is a precursor of the non-beta-amyloid component (Abeta) of Alzheimer’s disease amyloid (NAC), which, together with amyloid Aβ, is a component of amyloid (senile) plaques and may initiate the aggregation of Aβ peptides [10,41].

Bacterial metabolites, such as short-chain fatty acids (SCFA), produced by the fermentation of polysaccharides (starch, starch-like, non-starch) affect the gut–brain axis by regulating the synthesis of 5-HT by enterochromaffin cells, which influence hormonal communication between the gut and the brain. 5-HT activates afferent nerve endings that transmit information to the CNS. SCFAs have the ability to cross the blood–brain barrier and are responsible for maintaining microglial homeostasis, which is essential for the proper functioning of nervous tissue. Numerous studies on cell cultures and animal models have confirmed that SCFAs ensure the tightness of the vascular endothelium of CNS cells, reduce inflammation, improve glucose utilization, and also affect the brain through direct humoral effects, indirect hormonal and immunological pathways, and neural pathways [42,43,44]. SCFAs mainly consist of acetic, propionic, and butyric acids (physiologically occurring in proportions of 60:25:15, respectively), as well as formic, valeric, and caproic (caprylic) acids. They are produced in the intestines during anaerobic fermentation, mainly of complex carbohydrates (non-starch polysaccharides, dietary fiber, resistant starch) and, to a small extent, proteins (proteins provide a greater variety of end products, including SCFA, amines, phenols, indoles, thiols, CO_2_, H_2_, and H_2_S, but many of them have toxic properties). Oligosaccharides (e.g., fructooligosaccharides, xylooligosaccharides), branched-chain amino acids (e.g., isobutyrate, isovalerate), intermediate fermentation metabolites (e.g., lactate, ethanol), and glycoprotein–mucin are produced by the intestinal mucosa [45]. The increased presence of SCFA depends mainly on the quality of the diet based on the supply of products containing, among others, dietary fiber (also known as dietary fiber) and resistant starch, which provides the most butyric acid. Their main sources are wholemeal products, wholemeal bread, coarse groats and dark pasta, bran, seeds, nuts, vegetables (beans, peas, broad beans, cabbage, lettuce, spinach, carrots, beetroot, tomatoes, potatoes), and fruit (currants, blackberries, blueberries, apples, pears, dried fruit) [46,47,48]. Dietary fiber and resistant starch are the main sources of propionic acid. In a properly balanced diet that provides the right proportions of protein, carbohydrates, fats, and dietary fiber and is additionally enriched with probiotics and prebiotics, the level of SCFAs, including butyric acid, is sufficient. The bacteria that produce butyric acid are primarily *Eubacterium* spp., *Clostridium* spp., *Fusobacetrium*, *Megasphaera elsdenii*, *Butyrivbrio* spp., and *Mitsuokella* multiacida [49]. The formation of SCFA is a mutualistic process in which previously produced butyric acid and propionic acid can be degraded to acetic acid by acetogenic bacteria, including *Acetobacterium*, *Acetogenium*, *Eubacterium* spp., and *Clostridium* spp. The resulting acetate can then be converted to butyrate and propionate if the number of butyric-acid-producing bacteria increases, i.e., by *Faecalibacterium prausnitzii* and *Roseburia* species. In addition, acetate produced by *Bacteroidetes thetaiotaomicron* may act as a substrate for butyrate synthesis by *Eubacterium rectale* [45].

Disorders in the development of the gut microbiome affect not only the functioning and development of the digestive system and immune systems but also the nervous system. The cross-talk between the gut microbiota and the brain may have a crucial impact on neurodegenerative disorders.

## 3. Dysbiosis and Bacterial Endotoxins: LPS, BF-LPS, and BFT

More and more experimental and clinical data confirm the key role of intestinal dysbiosis in neurodegenerative diseases, including AD. According to current knowledge, intestinal dysbiosis may be a cause of inflammation in the CNS, especially because it has been shown that in people with cerebral amyloidosis pro-inflammatory bacteria predominate in the intestine, while anti-inflammatory bacteria are reduced. Changes in the gut microbiota are associated with elevated levels of pro-inflammatory cytokines, oxidative stress, and neurotoxicity in the brain tissue of patients with AD [50,51]. Dysbiosis is an imbalance in the gut microbiota consisting of quantitative changes in individual bacterial species and/or qualitative changes resulting from a reduction in species diversity, which ultimately leads to the malfunctioning of the bacterial ecosystem. Dysbiosis can be caused by various factors, such as excessive alcohol consumption, antibiotics, non-steroidal anti-inflammatory drugs, processed foods, and a diet low in fiber. The effect of dysbiosis is increased intestinal permeability, which leads to the release of endotoxin–lipopolysaccharide (LPS) into the blood, which is considered the strongest pro-inflammatory, neurotoxic glycolipid derived from anaerobic Gram-negative bacteria. In addition, the glycolipid subtype LPS–*Bacteroides fragilis* (BF-LPS) and the proteolytic, zinc-dependent metalloprotein toxin *Bacteroides fragilis toxin* (BFT), i.e., fragilisin, secreted by *Bacteroides fragilis*, are particularly neurotoxic and pro-inflammatory. BF-LPS and BFT, as well as EC-LPS (produced by *Escherichia coli*), disrupt the cell barrier of the intestinal epithelium and break down intercellular proteins, such as E-cadherin, leading to increased intestinal barrier permeability (these barriers become more permeable with aging). In addition, endotoxins promote the production of the pro-inflammatory nuclear transcription factor NF kappa B (NF-κB) formed from the combination of p50/RelA proteins (p50/p65 heterodimer), triggering a strong pro-inflammatory response and disrupting homeostasis in neuronal–glial cells [52,53,54]. LPS, BF-LPS, and BFT have been shown to be involved in neurological diseases, including inflammatory neurodegeneration, and to contribute to cognitive impairment. The mechanism of action of LPS involves destructive effects on presynaptic proteins (NRXN-1, SNAP-25, SYN-2), postsynaptic proteins (NLGN and SHANK3), and neuronal intermediate filament protein, which are involved in the structure and function of synapses, and by creating specific changes in synaptic transmission of neurons, contributing to cognitive deficits characteristic of the neurodegenerative process in AD [55,56]. In addition, circulating LPS and bacterial amyloid fibers synthesized by certain bacteria (*Escherichia coli*, *Salmonella enterica*, *Salmonella typhimurium*, *Bacillus subtilis*, *Mycobacterium tuberculosis*, *Staphylococcus aureus*) [57] activate receptors in intestinal epithelial cells, including Toll-like receptors (TLR4); with the participation of the co-receptor MD-2 protein and the receptor for advanced glycation end products (RAGE), they enhance pro-inflammatory signaling, leading to chronic nerve inflammation and thus progressive neurodegeneration, particularly in the hippocampus (one of the regions of the brain most affected by Alzheimer’s disease) [58]. RAGE is a receptor from the immunoglobulin family that binds advanced glycation end products (AGEs), which are physiologically produced in aging cells and found in heat-treated foods (frying, baking, grilling) [8]. The binding of amyloid Aβ to RAGE, which transports Aβ through the dysfunctional intestinal endothelium barrier into the bloodstream and then through the blood–brain barrier into the brain, accelerates the accumulation of Aβ in neurons, microglia, and vascular cells, leading to biochemical changes in memory neurotransmitters (including glutamate and acetylcholine), which accelerates the harmful effects on neuronal and synaptic functions and, consequently, exacerbates learning and spatial memory impairment [59,60]. The abundance and secretion of lipopolysaccharides from Gram-negative bacteria can be stimulated by various factors (medications, lifestyle, age), as well as substances found in the diet, including high fat and cholesterol intake and insufficient dietary fiber. It has been suggested that a diet containing adequate amounts of dietary fiber may play an important role in regulating the abundance of *B. fragilis* and the BF-LPS it secretes [53]. The European Food Safety Authority (EFSA) guidelines recommend a dietary fiber intake of 25 g/day for adults and 10–21 g/day for children (depending on age) [61].

An in vitro experiment with *Escherichia coli* endotoxins and Aβ peptide showed an increase in fibrillogenesis, indicating that bacterial endotoxins may play a key role in the pathogenesis of AD [50]. It is worth mentioning here another role of the Aβ peptide, which has a protective antimicrobial function, participates in the innate immune response, and fights infections in the brain [62]. It has been found that β-amyloid fibrils inhibit the adhesion of pathogens to host cells and mediate agglutination and entrapment of microorganisms (they bind to carbohydrates in the cell wall of microorganisms via a heparin-binding domain) [62,63]. Aβ fibrillation activates neuroinflammatory pathways that help fight infection and remove β-amyloid or pathogen deposits. Studies conducted by Kumar et al. [62] in the brains of transgenic 5XFAD mice infected with *Salmonella typhimurium* resulted in rapid synthesis and accelerated deposition of β-amyloid, indicating a protective role for β-amyloid in innate immunity to intestinal pathogens or other inflammatory stimuli that stimulate amyloidosis. However, given the antimicrobial properties of β-amyloid, it remains unclear why cells are unable to eliminate β-amyloid/pathogen deposits in individuals with AD. The hypothesis of antimicrobial protection by β-amyloid in AD assumes that chronic activation of this pathway in the brain ultimately leads to chronic inflammation [64] and increased β-amyloid deposition, neurodegeneration, and disease progression [62,63].

As a result of dysbiosis, the growth of pro-inflammatory strains, and the action of bacterial amyloid, inflammation develops in the CNS, which predisposes to quicker onset of AD symptoms. In older people, the role of the gut microbiota in amyloid formation becomes more serious because small molecules easily penetrate the more permeable intestinal epithelium and the blood–brain barrier [65].

## 4. Diet for the Gut Microbiome and the Brain

Proper nutrition, including a balanced diet, avoiding stimulants, and a healthy lifestyle help maintain balance in the gut microbiota and the proper functioning of the intestinal barrier and the gut–brain axis and thus reduce the risk and alleviate the symptoms of neurodegenerative diseases, including AD. The most important factor influencing the composition of the gut microbiota is diet, in which the type and composition of food are important. The best diet for the gut microbiome should be varied, especially in terms of fruit and vegetables, as eating them will increase the diversity and abundance of beneficial bacteria, such as *Bifidobacterium* and *Lactobacillus*, and reduce the number of harmful bacteria, such as *Enterococcus* spp. and *Escherichia coli* [66]. Studies show that eating 30 different foods per week, compared to less than 10, is associated with significantly greater potential of the gut microbiome, and that dietary diversity each day also influences the diversity and stability of the microbiome. Increased consumption of specific food components, such as kefir (increases the abundance of *Lactobacillus reuteri*, *Eubacterium plexicaudatum*, *Bifidobacterium pseudolongum*, *Parabacteroides goldsteinii*, *Bacteroides intestinalis*, *Anaerotruncus unclassified*, and *Alistipes unclassified*), probiotic milk, grapes, and pomegranate, can modify the gut microbiota to a healthy enterotype [10]. These food components influence the production of metabolites beneficial to the body (e.g., SCFA, indole-3-propionic acid, urolithin A) and may prevent the risk of neurodegenerative diseases by reducing inflammation of the nervous system, oxidative stress, neurotransmitter dysfunction, and neuron death [10]. It is believed that a balanced diet that is healthy for the gut and the brain should primarily include legumes, pseudocereals (quinoa, amaranth, millet, khorasan, sorghum, teff, barley), fresh herbs and spices, fruits and vegetables, whole grain products, nuts, fish, and low-fat dairy products, as well as dietary fiber, a component of which is resistant starch, which is not subject to enzymatic digestion and passes into the large intestine in an unchanged form, where it undergoes decomposition. Consuming plant-based diets rich in soluble fiber found in apples, strawberries, pears, apricots, berries, currants, plums, dried beans, citrus fruits, potatoes, barley (flakes, bran, groats), oats (bran, flakes), brown rice, parsley, and carrots prevents dementia in AD. This view is confirmed by the study by Cuervo-Zanatta et al. [67] conducted on a transgenic mouse model with Alzheimer’s disease. The study showed that soluble fiber intake modulates the composition of the gut microbiota and restores the normal production of SCFAs by gut bacteria, with higher butyrate and lower propionate production (intestinal dysbiosis was associated with intestinal damage and high propionate levels). The study found an increase in butyrate-producing bacteria (e.g., *Bacteroides*, *Clostridiales*, *Oscillospira*, *Moryella*, *Dehalobacterium*) and confirmed that increased butyrate concentrations were associated with improved cognitive abilities in the mice studied. It is concluded that the neuroprotective effect of soluble fiber is associated with the normal production of butyrate/propionate by intestinal bacteria [67].

One of the dietary fiber fractions, the so-called insoluble fraction, is resistant starch, which is found in all products containing degraded amylose (wheat bran, legume seeds, whole or partially ground grains and seeds, green bananas, cooked and cooled potatoes, pasta and rice, stale bread). Resistant starch is fermented in the large intestine with the participation of probiotic lactic acid bacteria, i.e., *Bifidobacterium* and *Lactobacillus*, which form the basis of healthy human intestinal microflora. As demonstrated in the publication by Deehan et al. [46], fiber from potatoes, corn, and tapioca had different effects on the abundance and diversity of bacteria. For example, tapioca consumption caused an increase in the abundance of bacteria from the *Porphyromonadaceae* family (*Parabacteroides distasonis*, *Parabacteroides* spp., *Faecalibacterium prausnitzii*, *Eisenbergiella* spp.). In contrast, an increase in the abundance of *Bifidobacterium adolescentis* was obtained after consumption of both corn and tapioca [46].

A diet rich in fermented foods produced using acetic acid bacteria (AAB) has a beneficial effect on human health [68,69]. Acetic acid bacteria belonging to *Acetobacter pasteurianus* are found in fruits (apples, bananas, grapes, oranges, papaya, peaches, pineapples, passion fruit, gooseberries, strawberries, melons) and vegetables (tomatoes) [70]. AAB has also been isolated from Arabica coffee beans, fermented cocoa beans, flowers, corn, pollen, sugar cane roots, and wild rice. The most well-known application of aerobic AAB is in the production of vinegar (wine vinegar is the most popular type of vinegar in Mediterranean countries) [71]. A fermented tea obtained from a symbiotic culture of acetic acid bacteria (*Komagataeibacter*, *Gluconobacter*, *Acetobacter* spp.), lactic acid bacteria, and yeast is a popular drink among many fermented food products. Kombucha is a source of many vitamins (C, B1, B2, B3, B6, B9, B12), enzymes, amino acids, nucleic acids, and organic acids (acetic, lactic, malic, oxalic, carbonic) and has antioxidant, antimicrobial, anti-inflammatory, and anti-cancer properties. Kombucha primarily contains live bacterial cultures, which give the drink its probiotic properties. In addition, the D-saccharic acid 1,4-lactone (DSL) and glucuronic acid (GlcUA) contained in the drink (with detoxifying and antioxidant properties) are considered key functional components of kombucha [68,72]. AAB bacteria, such as *Acetobacter lambici*, *Acetobacter cerevisiae*, *Acetobacter pasteurianus*, and *Gluconobacter cerevisiae*, are used to produce sour lambic beers, which are becoming increasingly popular around the world thanks to their refreshing acidity and fruity notes [73]. Research by Fukami et al. conducted on 66 healthy participants aged 50–69 showed that *Acetobacter malorum* NCI 1683 (S24) derived from fermented milk can improve cognitive function after only 8 weeks of consumption, not only at high doses (400 mg/day) but also a low dose (111 mg/day). The authors of the study suggest that intake of acetic acid bacteria from fermented foods is effective in improving cognitive function in healthy middle-aged and elderly people. Foods containing acetic acid bacteria may have a beneficial effect on cognitive function in older people, slowing down the natural and inevitable biological and mental aging process of the human body [74].

Including polyphenol-rich foods in the diet promotes the growth of beneficial microorganisms, such as *Bifidobacterium* and *Lactobacillus*, and reduces the number of harmful bacteria, including *Staphylococcus aureus*, *Salmonella typhimurium*, and *Clostridium* spp. In addition to their anti-inflammatory properties, polyphenols and their metabolites prevent cognitive decline and may prevent neurodegenerative diseases through their anti-glycation effects on the AGEs–RAGE axis and by regulating the microflora–gut–brain axis [75]. Recent studies show that foods rich in bioactive polyphenols found in seaweed can influence a healthy gut microbiota enterotype and reduce the risk and alleviate the symptoms of neurodegenerative diseases. It turns out that regular consumption of seaweed, which is rich in bioactive polyphenols, such as unsaturated mannuronate oligosaccharide, e.g., polymannuronic acid (PM), which is a component of alginate found in the cell walls of brown seaweed, has neuroprotective effects, possibly by alleviating inflammation in the intestines, brain, and systemic circulation, as well as by strengthening the integrity of the blood–brain barrier and the intestinal barrier. Furthermore, PM has been shown in a mouse model to lead to significantly reduced production and deposition of Aβ_1–42_. Another compound, phlorotannin, found in brown algae, exhibits the strongest anti-aggregatory effect against Aβ_1–42_ (inhibiting it by over 90%) and anti-apoptotic effects in preventing Aβ_1–42_-induced cell death (reducing the rate of apoptosis by almost threefold in the rat model studied) [10]. Sodium oligomannate (GV-971), an oligosaccharide found in brown algae, administered orally at a dose of 450 mg twice daily for 36 weeks in a phase III clinical trial involving patients with AD (mild to moderate) significantly improved the cognitive function of these patients [10]. Furthermore, in mouse model studies conducted by Bosch et al. [76], GV-971 significantly reduced cerebral amyloidosis, mainly in males. Changes in the composition of the gut microbiota and a significant effect on the metabolism of the microbiome were also observed, particularly through increased production of amino acids and an effect on the tryptophan pathway. A reduction in the amount of pro-inflammatory cytokines and chemicals that reduce neuroinflammation was also found [76]. In another study, Gong et al. [77] noted that GV-971 can activate the gut–brain axis by activating enteroendocrine cells and afferent fibers of the vagus nerve (through an increase in serotonin and cholecystokinin), thereby improving cognitive function. GV-971 has recently been approved for the treatment of patients with AD in China [77].

Increasing attention is being paid to the fact that an unhealthy diet in people following a Western diet has a significant negative impact on the functioning of the gut microbiota. Consuming foods rich in saturated fats and trans fats, such as margarine, fast food, and sweet and salty snacks, as well as excessive amounts of refined sugars, salt, and red meat and low fiber content, contributes to a reduction in the diversity, abundance, and stability of the gut microbiota [78]. Studies have shown that a Western diet (characterized by high consumption of saturated fats, sugar, and salt and low consumption of fiber, vegetables, and fruit) causes a significant reduction in microorganisms of the genus *Bifidobacterium*, *Bacteroides*, and *Prevotella* and bacteria producing butyric acid, mainly of the genus *Clostridium*, *Eubacterium*, and *Fusobacterium*, with a simultaneous increase in Firmicutes (Bacillota) [79].

According to many studies, the best dietary pattern for brain function is the Mediterranean diet, a modified ketogenic–Mediterranean diet, DASH, and MIND, which may prevent and slow the development of AD [13].

### 4.1. The Mediterranean Diet

The Mediterranean diet (MedDiet) is considered one of the healthiest diets and is mainly associated with the prevention of cardiovascular disease, diabetes, and cancer. Thanks to its high content of bioactive ingredients, this dietary model also has a preventive effect on the nervous system [80]. The MedDiet is characterized by high consumption of fruit, vegetables (especially green, leafy vegetables), nuts and seeds (and oil extracted from them), and legumes, which are a source of omega-3 polyunsaturated fatty acids, especially alpha-linolenic acid (ALA), which is converted in the body in very small amounts (about 5%) to eicosapentaenoic acid (EPA) and about 1% to docosahexaenoic acid (DHA). To supplement EPA and DHA deficiencies, fatty fish, such as salmon, tuna, mackerel, and herring, as well as fish oils, which are their main source, should be included in the diet. Another important health factor is linoleic acid (LA), an essential unsaturated fatty acid belonging to the omega-6 family of fatty acids, which must be supplied to the body through food. It is found in large quantities in vegetable oils, nuts, and seeds and in smaller quantities in cereals, meat, eggs, dairy products, and legumes [11]. In addition, the MedDiet recommends eating fresh fish (salmon, tuna, mackerel, trout, sardines), seafood and fish oil, and olive oil (used as a basic fat), which is rich in monounsaturated fatty acids. This diet assumes moderate consumption of milk and dairy products (cheese, yogurt, buttermilk, cottage cheese), eggs, and white meat (poultry, rabbit meat) and low or moderate consumption of alcohol (wine, beer) with meals [8,9]. The MedDiet recommendations include low consumption of red meat and meat products and very low consumption of processed foods (fried foods, salty snacks, sweets, high-fat dairy products, sugar-sweetened non-alcoholic beverages) [81] (Table 1).

Fruits and vegetables have protective properties due to their high content of vitamins, minerals, dietary fiber, plant sterols, flavonoids, and other antioxidants, which have a beneficial effect on the growth of *Bifidobacterium*, *Lactobacillus*, and *Bacterioidetes* [82]. The presence of polyphenols in the diet has a beneficial effect on the bacteria colonizing the intestines, e.g., catechin significantly inhibits the growth of *Clostridium histolyticum* and increases the growth of *Escherichia coli* and members of the *Clostridium coccoides–Eubacterium rectale* group, while the number of *Bifidobacterium* and *Lactobacillus* spp. remains relatively unchanged. In turn, compounds like isoflavones (daidzein), flavanols (quercetin and kaempferol), flavones (naringenin and ixoxanthumol), and flavan-3-ols (catechin and epicatechin) are metabolized by the presence of bacteria of the *Clostridium and Eubacterium* genera [83]. An experimental study in a rat model showed that a diet supplemented with polyphenols (proanthocyanidin) from red wine changed the composition of fecal bacteria in favor of *Bacteroides*, *Lactobacillus*, and *Bifidobacterium*, rather than *Clostridium and Propionibacterium* spp. [84]. Similarly, resveratrol, commonly found in grapes, increased the number of *Bifidobacterium* and *Lactobacillus* in feces [85]. Interesting results were obtained in people who consumed moderate amounts of red wine for 4–5 weeks. This study observed an increase in the overall diversity of bacteria, including *Proteobacteria*, *Fusobacteria*, *Firmicutes*, and *Bacteroidetes* and an increase in important health-promoting bacteria with anti-inflammatory properties that produce butyrate, such as *Faecalibacterium prausnitzii* and *Akkermansia* spp. These results suggest that wine may modulate the intestinal bacterial ecosystem and also have a positive effect on brain health, preventing AD [86]. Alcohol consumption is neither a necessary nor a recommended part of the diet. It is believed that there is no safe dose of alcohol. Even small amounts can have a negative impact on health.

It is worth adding information presented by Clark et al. [87] concerning the relationship between adherence to the MedDiet and increased SCFA concentrations in feces, as well as the abundance of Bacteroides and Firmicutes (Bacillota), whose abundance is an important indicator of human health. As it turns out, beneficial changes in the gut microbiota and SCFA production by bacteria can be achieved in a short period of time on the MedDiet, even within just 4 days. However, as noted, this change is temporary, as after discontinuing the MedDiet, the microbiota returns to its previous profile after a few weeks. In another cross-sectional study conducted by Rosés et al. [88] in 360 Spanish adults following the MedDiet (legumes, vegetables, fruit, olive oil, nuts, fiber), the greatest increase was observed in *Bifidobacterium animalis*, belonging to the *Bifidobacterium* genus. In addition, the abundance of other SCFA-producing bacteria, such as *Roseburia faecis*, *Ruminococcus bromii*, and *Oscillospira (Flavonifractor) plautii*, was also increased. The study analyzed the type of food that influences the growth of specific bacteria populations. Namely, it was shown that fiber intake increased the growth of SCFA-producing bacteria, such as *Oscillibacter valericigenes*, *Oscillospira* (*Flavonifractor) plautii*, and *Roseburia faecis.* It was also noted that the abundance of *Roseburia faecis* increases after the consumption of fruits and nuts. Legumes, on the other hand, enhance the production of *Ruminococcus bromii*, while vegetables increase the population of *Butyricicoccus pullicaecorum.* Nut consumption also has a beneficial effect on the growth of *Papillibacter cinnamivorans*. Another interesting study was conducted by Hoscheidt et al. [89], who observed clear differences between a diet based on the Mediterranean diet and a Western diet in middle-aged adults (45–65 years) with normal cognitive function (NC) and mild cognitive impairment (MCI) and AD biomarkers in cerebrospinal fluid. In the NC group on the MedDiet, a decrease in the concentration of Aβ_40_ and an increase in the Aβ_42/40_ ratio were observed, indicating a reduced risk of AD. In addition, increased cerebral perfusion and improved cognitive function were found in the NC group on the MedDiet. In contrast, on the Western diet, the NC group showed an increase in Aβ_40_ and a decrease in the Aβ_42 /40_ ratio_,_ which is associated with an increased risk of AD. The study presents very interesting data on the Western diet, which may have beneficial effects for adults with MCI. In MCI participants on a Western diet, an increase in the Aβ_42/40_ ratio and reduced t-tau protein concentration and Aβ_42_/t-tau ratio were observed, indicating a lower risk of AD. The opposite results were obtained in individuals with MCI following the MedDiet, who had increased t-tau concentrations and a low Aβ_42_/t-tau ratio, indicating a higher risk of developing AD [89]. In another study, magnetic resonance imaging (MRI) in healthy older adults aged >60 years found that lower adherence to the MedDiet (higher consumption of red and processed meat and lower consumption of unrefined cereals, fruit, vegetables, legumes, nuts, and olive oil) was associated with increased atrophy in brain areas specific to Alzheimer’s disease (hippocampal and dentate gyrus volumes) and also characterized by poorer memory and learning [90].

It is believed that a diet based on the Mediterranean diet and rich in fruit and vegetables can modulate and slow down the progression of AD. The particularly high content of fiber and polyphenols, due to their antioxidant, anti-inflammatory, and anti-apoptotic properties, is a factor contributing to the preservation of microbiota diversity and may therefore strengthen cognitive functions and prevent dementia.

Summary:



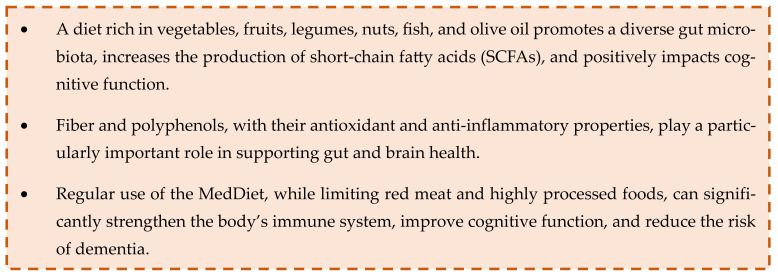



### 4.2. The Dietary Approaches to Stop Hypertension Diet

Dietary Approaches to Stop Hypertension (DASH) was originally developed to reduce the risk of developing high blood pressure and cardiovascular disease. Like the MedDiet, the DASH diet emphasizes high consumption of vegetables, fruits, nuts, whole grains, legumes, seeds, and healthy fats and moderate consumption of fish and poultry and low-fat dairy products (Table 1). The DASH diet recommends limiting red meat, foods containing saturated fatty acids, and salt (sodium content < 2.4 g/day) and avoiding highly processed foods, sweets, sugary drinks, and alcohol [12]. Like the MedDiet, the DASH diet is a source of antioxidants, B vitamins, polyphenols, PUFA (polyunsaturated fatty acids), MUFA (monounsaturated fatty acids), and DHA. The DASH diet is believed to protect the brain and may prevent Alzheimer’s disease due to its high content of neuroprotective bioactive compounds, such as omega-3 fatty acids, antioxidants, and polyphenols. Long-term adherence to this diet has positive effects on cognitive function and psychomotor activity [12]. One of the ENCORE (Exercise and Nutrition Interventions for Cardiovascular Health) studies [91] showed an improvement in cognitive function and memory after just 4 months of following a calorie-restricted DASH diet combined with aerobic exercise in overweight/obese adults with hypertension (i.e., prehypertension and in the first stage of hypertension). It was noted that this improvement was due to weight loss and improved cardiorespiratory fitness, as high blood pressure increases the risk of stroke, dementia, and neurocognitive dysfunction [91]. In contrast, another 6-month randomized clinical trial, ENLIGHTEN (Exercise and Nutritional Interventions for Neurocognitive Health Enhancement), involving adults (aged > 55) with mild cognitive impairment who were on the DASH diet did not show significant improvement in cognitive function, although greater benefits were observed with the DASH diet combined with aerobic exercise [92].

An analysis of evidence assessing the relationship between the DASH diet and cognitive function, dementia, or Alzheimer’s disease was conducted by van den Brink et al. [13]. This analysis shows that the more participants adhered to the DASH diet, the lower their risk of dementia (approx. 39%) and the better their cognitive function (verbal, episodic, and semantic memory). The authors cite two publications in which several years of observation (from 4.1 years to 10.6 years) of Swedish and American participants showed no link between the DASH diet and improved cognitive function. In another study, CARDIA (Coronary Artery Risk Development in Young Adults) [93] compared two diets, MedDiet and DASH. The study showed that high adherence to the MedDiet in middle age is associated with better cognitive abilities, while adherence to the DASH diet does not show such properties. According to the authors of the study, one reason may be that the DASH diet does not include moderate alcohol consumption, whereas the MedDiet and MIND diets include alcohol. It appears that moderate alcohol consumption, as part of a healthy dietary pattern, may be important for brain health [93]. Available evidence suggests that both the MedDiet and DASH dietary patterns may provide protection against neurodegeneration during aging, with the MedDiet having significantly greater benefits for brain health, although a randomized Spanish study (PREDIMED-NAVARRA [PREvención con DIeta MEDiterránea-NAVARRA]) showed a small beneficial effect of adhering to the MedDiet for 4–6 years on cognitive function in older adults aged 74.6 ± 5.7 years with normal cognitive function but at high risk of cardiovascular disease [94]. Other authors, such as McGratta et al. [95], argue that the vast amount of bioactive compounds and nutrients (e.g., omega-3 fatty acids, antioxidants, polyphenols) found in both the MedDiet and DASH diets have a significant impact on reducing neuroinflammatory processes and may thus limit the development of neurodegenerative diseases. For example, polyphenols modulate the intestinal ecosystem by inhibiting the growth of pathogenic bacteria and stimulating the growth of beneficial species associated with the thickness of the intestinal mucus layer, such as *Lactobacillus* spp., *Bifidobacterium* spp., and *Akkermansia muciniphila* [96]. In addition, the DASH diet, rich in whole grains, fruits, and vegetables and containing more fiber than the standard American diet, can contribute to changes in the gut microbiota and improve the condition of the intestinal mucosa in just 2–3 weeks of following the DASH diet. As shown by Carson et al. [97], in people on the DASH diet, the abundance of the gut microbiota undergoes significant changes. A reduced relative abundance of *Fusobacterium*, *Porphyromonas*, *Bacteroides*, *Bifidobacterium*, and *Succinivibrio* and an increased abundance of *Clostridium*, *Ruminococcus*, and *Lactobacillus* are contrary to individuals on a typical American diet.

The benefits of following the DASH diet are manifold, including lower blood pressure, powerful anti-inflammatory effects, which reduce the risk of many lifestyle diseases, improved lipid parameters, support for diabetes and insulin resistance, and improved intestine function (high fiber content has a beneficial effect on the digestive system).

Summary:



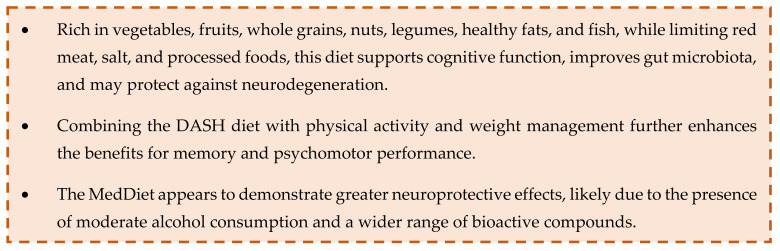



### 4.3. The Mediterranean–DASH Intervention for Neurodegenerative Delay Diet

A dietary model aimed at preventing dementia and increasing neuroprotection should regulate modifiable risk factors for cardiovascular disease (hypertension, diabetes, dyslipidemia, overweight and obesity, low physical activity) and slow down dementia and neurodegenerative processes. Healthy eating habits should be introduced as early as possible, especially in middle-aged people, long before the first symptoms of dementia appear [95]. The concept of the Mediterranean–DASH Intervention for Neurodegenerative Delay diet (MIND) is a combination of two dietary models, MedDiet and DASH, recognized as the healthiest diets in the world. The advantage of the MIND diet is that there are no strict restrictions on the products used, so it is easier to follow than other diets. Compared to MedDiet and DASH, the MIND diet further specifies fruit consumption, as it introduces berries in place of fruit as a general category. The components of the MIND diet have been divided into two groups of products—recommended, based mainly on plant products, which have a positive effect on cognitive brain function, including green leafy vegetables, other vegetables, nuts, berries, beans/legumes, whole grains, fish, poultry, olive oil, and wine—and a group of non-recommended products, including red meat, fried and fast foods, pastries and sweets, butter and margarine (no more than one teaspoon per day), and cheese (less than once a week), as they contain saturated fats and trans fats [12]. Berries and poultry should be consumed at least twice a week and fish at least once a week. In addition, as in the MedDiet, one glass of wine per day is acceptable [98]. The MIND diet recommends consuming (a) at least six servings of green leafy vegetables per week; (b) at least three servings of whole grains per week; (c) one serving of other vegetables per day; (d) nuts at least five times per week; and (e) consumption more than three times per week of legumes (i.e., beans, chickpeas, lentils, peas). Olive oil should be the primary source of fat [98,99] (Table 1).

Green leafy vegetables are rich in nutrients, including vitamin E, folates, β-carotene, lutein, zeaxanthin, and flavonoids, which contribute to better brain function and protect neurons from damage associated with oxidative stress caused by free radicals [100]. Consuming berries increases neurogenesis and insulin-like growth factor-1 (IGF-1) signaling and slows down neuron aging by reducing oxidative stress [101]. Consumption of leafy vegetables (lettuce, spinach, broccoli, kale, cabbage), which are the main source of folates (natural form of vitamin B9), has an effect on lowering homocysteine levels, which may be associated with a lower risk of developing dementia in Alzheimer’s disease. Folic acid supplementation for 3 years in study participants resulted in a decrease in plasma homocysteine levels, as well as improved memory and information processing speed and improved sensorimotor integration [102].

The assessment of the relationship between regular adherence to the MIND diet and cognitive function, dementia, or Alzheimer’s disease was presented by van den Brink et al. [13]. The study showed that greater adherence to the MIND diet was significantly associated with a smaller decline in cognitive function after 4.7 years of follow-up in all five cognitive domains measured, i.e., episodic memory, semantic memory, visuospatial memory, perceptual speed, and working memory. In another study, after 12 years of follow-up, it was confirmed that greater adherence to the MIND diet was associated with a 19% lower likelihood of mild cognitive impairment and dementia [103]. Autopsy results also showed that people who followed the MIND diet had fewer changes in brain structure. In addition, earlier observations of these patients confirmed better improvement in cognitive function and a slower decline compared to patients who did not follow this dietary model [100]. Evidence suggests that following the MIND diet can reduce the risk of dementia and improve cognitive function in older adults and prevent or slow the onset of AD. The MIND diet is more beneficial than other plant-rich diets, such as the MedDiet, DASH, Pro-Vegetarian, and Baltic Sea diets [104]. Following the MIND diet and modifying the gut microbiome, e.g., by administering probiotics with *Lactobacillus* and *Bifidobacterium* (especially *Bifidobacterium breve A1* and *Lactobacillus plantarum C29* strains) and prebiotics (inulin, fructooligosaccharides), may prevent cognitive decline and dementia [105]. Natural sources of prebiotics include garlic, onions, leeks, asparagus, peas, beans, Jerusalem artichokes, bananas, and whole grain cereals. One of the best sources of prebiotics is fermented fruits and vegetables.

Therefore, maintaining a balanced gut microbiota through proper nutrition may prevent or delay the process of neurodegeneration in Alzheimer’s disease.

Summary:



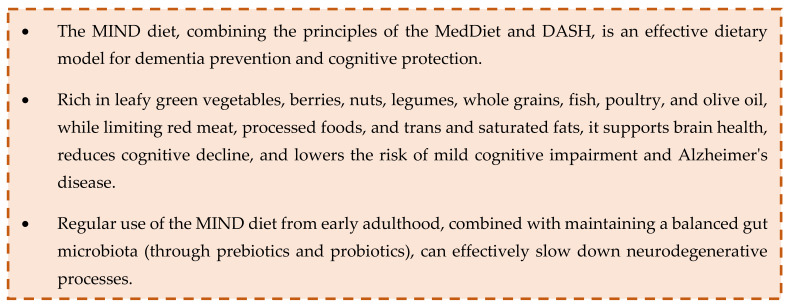



### 4.4. The Ketogenic Diet

The ketogenic diet (KD) is another dietary approach used in the prevention and treatment of neurodegenerative diseases, including Alzheimer’s disease, Parkinson’s disease, epilepsy, depression, autism, and traumatic brain injury [14,106]. The KD is a high-fat, low-carbohydrate diet with adequate protein intake, leading to increased production of ketone bodies, such as acetooctanoic acid, β-hydroxybutyric acid, and acetone. The main goal of this diet is to induce ketosis in the body and use ketone bodies as an alternative source of energy for the central nervous system instead of glucose [107]. Ketone bodies are synthesized in the liver as a result of mitochondrial β-oxidation of fatty acids with the production of acetyl-CoA [108]. Acetyl-CoA molecules can be used in the Krebs cycle or undergo condensation to acetoacetyl-CoA, which, under the influence of subsequent enzymatic reactions, is converted to acetoacetate, from which acetone is formed as a result of non-enzymatic decarboxylation and, in a reversible reaction, catalyzed by D-β-hydroxybutyrate dehydrogenase, through which β-hydroxybutyric acid is formed. A significant increase in the concentration of acetoacetate and β-hydroxybutyric acid in the blood crosses the blood–brain barrier and enters the brain, providing energy for neurons [109]. It has been proven that ketone bodies have a beneficial effect on the functioning of the nervous system and a neuroprotective effect on aging brain cells and improve circulation in the blood vessels of the brain. An example is a study conducted on a cell culture corresponding to the Alzheimer’s disease model, which showed that β-hydroxybutyric acid incubated with a β-amyloid fragment reduced cell mortality and increased the number of dendrites and the length of neurons [109].

Currently, there are many variations of the ketogenic diet that differ in the proportions of ingredients, allowing the diet to be tailored to various clinical disorders, such as diabetes, Parkinson’s disease, Alzheimer’s disease, autism, and epilepsy. The classic ketogenic diet consists of a 4:1 or 3:1 ratio of fat to protein and carbohydrates. For example, a 4:1 ratio (i.e., 4 g of fat for every 1 g of protein and carbohydrates) provides 90% of energy from fat, 7% from protein, and 3% from carbohydrates [107,110]. High-quality fats are found in nuts, avocados, olive oil, rapeseed oil, vegetable oils, and fish. The KD diet includes meat (chicken and turkey breast, pork loin, beef steak), fatty fish (herring, salmon, tuna, mackerel), shrimp, mussels, eggs, butter, cream, cheese, nuts, and non-starchy vegetables (cucumber, tomato, pepper, courgette, aubergine, cauliflower, broccoli, lettuce, spinach, kale) and fruits in small quantity with a low glycemic index (berries, apples, oranges, grapefruit, watermelon, kiwi). Alcohol and cereal products (rice, groats, pasta) are not recommended [110]. Following the KD diet is not easy and requires constant medical supervision. Restricting the consumption of carbohydrates contained in fiber-rich foods (e.g., legumes, whole grains), which are essential for the intestinal bacterial ecosystem, can be harmful to overall health. Therefore, to ensure that the diet is followed correctly, ketone body concentrations in the urine or blood should be monitored. In addition, it is recommended to check glucose, albumin, total protein, total cholesterol, triglycerides, and creatinine levels once every 3 months. Once a year, an ultrasound examination of the kidneys, bone density, carnitine, selenium, and an electrocardiogram should be performed, which are important in preventing long-term effects, such as kidney stones, osteoporosis, hyperlipidemia, carnitine deficiency, and cardiomyopathy [111].

The results of studies on the effect of KD on the functioning of the gastrointestinal tract, including the composition of the gut microbiota, its diversity, and the abundance of bacterial species, and on cognitive functions are inconclusive. In an experimental study conducted by Park et al. [112] using a rat model injected with β-amyloid, it was shown that the number and type of intestinal bacteria depended on the type of diet provided for 8 weeks. It was found that bacterial species diversity was higher on a high-carbohydrate diet than on a KD diet. In addition, intermittent fasting resulted in a decrease in the number of bacteria of the order *Clostridales*, among others, and an increase in the number of *Lactobacillales.* In contrast, the KD diet resulted in an increase in the number of *Proteobacteria*, especially *Enterobacteriales* (which are relatively harmful bacteria), compared to the other diets. Furthermore, both intermittent fasting and high-carbohydrate diets, but not the KD, reduced amyloid-β deposition in the hippocampus and improved memory function compared to the control group of rats on a normal diet (28% fat). In summary, the results of this experiment suggest that the KD diet does not improve cognitive function and reduces bacterial diversity, while intermittent fasting and a high-carbohydrate diet rich in starch may be beneficial for people with dementia. Different results in an experimental study on a mouse model fed a KD diet for 16 weeks were obtained by Ma et al. [113]. These authors reported an improvement in neurovascular disorders, suggesting a lower risk of developing Alzheimer’s disease. They also noted beneficial effects associated with changes in the composition of the gut microbiota. Namely, an increasing number of beneficial bacteria *Akkermansia muciniphila* and *Lactobacillus*, which have the ability to produce SCFAs, were found. They also found a reduction in pro-inflammatory bacteria, such as *Desulfovibrio* and *Turicibacter* [113,114]. It is known that KD reduces bacterial species diversity due to the low carbohydrate content (mainly dietary fiber) in the diet [113]. A low content of complex carbohydrates in the diet leads to a reduction in the number of *Bifidobacterium*, which have a beneficial effect on the human microbiota, and an increase in the number of *Escherichia coli*, which can have a harmful effect on the intestinal barrier. Studies conducted in people with MCI have shown an adverse effect of the KD diet on the gut microbiome, as this diet reduced the diversity of the bacterial flora, including an increase in the number of *Enterobacteriaceae* and a decrease in the number of beneficial bacteria from the *Bifidobacteriacea* family (of the *Actinobacteria* type). However, it has been shown that the KD diet restores the balance in the intestines by increasing the number of beneficial bacteria, such as *Akkermansia muciphila*, a species of bacteria that is capable of breaking down mucin (a component of the intestinal mucosa) and converting it into short-chain fatty acids (SCFA), including acetate. Acetate is used by other beneficial bacteria belonging to the Firmicutes (Bacillota) type to produce butyrate, an essential source of energy for the cells lining the intestine [115]. *A*. *muciphila*, through the constant breakdown of mucin, stimulates its continuous production, enabling the maintenance of the appropriate thickness of the mucous membrane. This bacterium supports the proper functioning of the intestinal barrier and improves the integrity of both the mucous membrane and epithelial cells, thus preventing pathogenic bacteria from penetrating deep into the tissues.

Although positive effects of the KD diet have been reported, negative effects on the gut microbiota, its diversity, and bacterial species abundance have also been observed. It turns out that long-term use of the KD may contribute to mucus degradation, which can lead to intestinal barrier dysfunction and the penetration of harmful microorganisms and their metabolites, which have adverse effects on the body. To prevent the adverse effects of the traditional KD, a more appealing version of this diet has been developed based on medium-chain triglycerides (MCT), where the main acids are acetic acid and decanoic acid [14]. In addition, this diet is easier to follow without drastic changes in food composition, as it is based on a normal diet supplemented with MCT. The results of a literature review conducted by Włodarek [116] indicate that people with AD with mild and moderate cognitive impairment who used a ketogenic MCT preparation showed significant improvement in cognitive function (memory, immediate and delayed logical memory) compared to baseline values. In another publication, Dyńka et al. [117] confirmed that supplementation with 30 g of MCT in patients with AD contributed to a doubling of ketone uptake in the brain and an increase in total brain energy metabolism (ketones produced from MCT compensated for glucose deficiency in the brains of patients with AD). It was noted that MCT supplementation significantly improves cognitive function, as an improvement in working memory was observed in 80% of patients with AD. The beneficial effects of kMCT (tricaprylin) in people with mild to moderate AD were also confirmed, with significant improvements in episodic memory, executive function, and speech, which was associated with increased uptake of ketone bodies in the brain, as seen in a PET [118]. Currently, the most popular and less restrictive variant of the classic ketogenic diet is the modified Atkins diet (MAD), consisting of 27% protein, 39% carbohydrates, and 34% total fat [14]. Supplementing KD with other exogenous ketogenic supplements, such as β-hydroxybutyric acid, or ketone esters e.g., (R)-3-hydroxybutyl(R)-3-hydroxybutyrate may alleviate aging processes, delay the onset of age-related diseases, prolong life, and improve learning and memory functions [119,120].

It should be noted that the restrictive rules of the KD diet, limiting the consumption of many food groups, may lead to metabolic disorders and deficiencies of important nutrients. For example, prolonged use may lead to deficiencies in protein, vitamins (B1, B6, C, folic acid), minerals (potassium, magnesium, calcium), and antioxidants (vitamins A and E, beta-carotene, lutein, lycopene, flavonoids, zinc, selenium, alpha-lipoic acid). In addition, a low dietary fiber intake may adversely affect the composition and function of the gut microbiota, leading to dysbiosis and the development of intestinal inflammation [106,121]. Therefore, to maintain proper gut microbiota function, high-carbohydrate vegetables and fruits can be replaced with low-carbohydrate products containing dietary fiber, e.g., green leafy vegetables (lettuce, chicory, celery leaves), cruciferous vegetables (broccoli, cauliflower, kale, Brussels sprouts), and other low-carbohydrate vegetables (asparagus, mushrooms, cucumber, green beans, courgette, peppers)**,** as well as berries (blueberries, raspberries, gooseberries, currants, cranberries), nuts, seeds, kernels, bran, and flaxseed. Additionally, supplementation with apple pectin, citrus fiber, acacia fiber, inulin or arabinogalactan, and polyphenols, as well as prebiotic fats, can be implemented in accordance with the principles of the Mediterranean–ketogenic diet [122].

Summary:



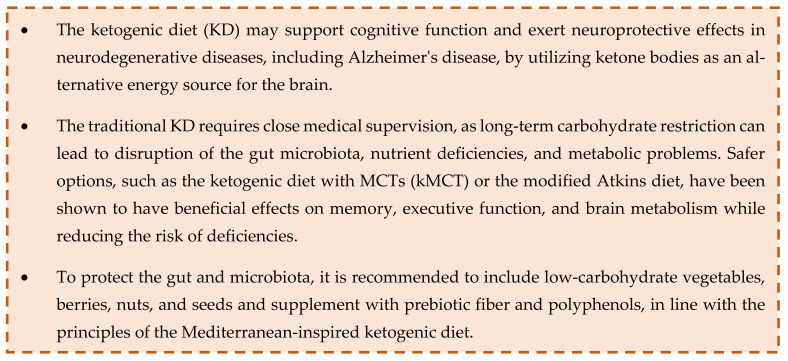



### 4.5. The Modified Mediterranean–Ketogenic Diet

The modified Mediterranean–ketogenic diet (MKD) comprises a combination of the principles of the classic ketogenic diet with features of the Mediterranean diet. This means giving up certain products characteristic of the Mediterranean diet and enriching it with some products from the ketogenic diet. The MKD diet recommends eating plenty of fatty fish (at least four times a week) and seafood, with additional fish oil supplementation on days when these are not consumed; high-quality olive oil instead of butter, oil, or lard; nuts and seeds; green leafy vegetables (lettuce, spinach, sorrel, beetroot, lamb’s lettuce, chicory, kale, rocket) as the main source of carbohydrates and other vegetables (aubergine, broccoli, cauliflower); eggs and full-fat dairy products; poultry occasionally; limiting or completely eliminating red meat, highly processed foods, and sugar-enriched products; significantly reducing the amount of carbohydrates consumed (pasta, traditional bread, certain fruits, and starchy legumes); and moderate alcohol consumption (wine).

It has been proven that the MKD diet has a beneficial effect on the human body by improving peripheral metabolism parameters. In plasma, a reduction in total cholesterol, triglycerides, LDL and VLDL cholesterol fractions, glucose, and HbA1c and an increase in HDL cholesterol have been demonstrated [15]. Experimental studies on a mouse model conducted by Park et al. [123] confirm the above statement. The study found, among other things, an increase in the number of *Parasuterella*, which improve the body’s metabolic functions by regulating bile acid (BA) concentrations. *Parasuterella* influences the expression of BA transporter genes in the ileum and BA synthesis genes in the liver, thereby maintaining BA concentration balance by reducing dietary cholesterol, which serves as a precursor for BA. An increase in the *Lactobacillus* population results in increased lactate production. MKD appears to provide all of the key dietary components necessary for maintaining proper gut microbiome function, thereby demonstrating a preventive effect against many diseases, including neurodegenerative diseases. Nagpal et al. [122] evaluated the effect of the MKD and American Heart Association (AHAD) diets on the concentration of AD biomarkers (Aβ_42_, Aβ_40_, total tau, tau-p181) in cerebrospinal fluid (CSF) and SCFA concentrations in feces and gut microbiota in elderly individuals (64.6 ± 6.4 years) with MC compared to individuals with normal cognitive function (CN). In that study, after 6 weeks on the MKD diet, changes in the gut microbiota were observed, e.g., the abundance of the *Actinobacteria* phylum, the *Bifidobacteriaceae* family, and the *Bifidobacterium* genus were significantly reduced in the MCI group compared to the CN control group and the MCI group on the AHAD diet. The target macronutrient composition (% of total calories) for the AHAD was 55–65% carbohydrates, 15–20% fat, and 20–30% protein (for the MKD, it was <10% carbohydrates, 60–65% fat, and 30–35% protein). At the same time, there was a decrease in lactate and acetate levels and a slight increase in propionate and butyrate in the MCI group on the MKD diet. The study demonstrated several patterns of positive and negative associations between AD biomarkers in CSF and gut microbiota and SCFA and diet. For example, the *Tenericutes* type, particularly in MCI participants on the MKD diet, was negatively correlated with Aβ_42_ in cerebrospinal fluid. In addition, the *Enterobacteriaceae* family was also negatively correlated with Aβ_42_ only in MCI individuals on the MKD diet. In contrast, a positive correlation was found between the *Lachnospiraceae* family and Aβ_42_ in both diets. A positive correlation was also found between the *Rikenellaeae* family and the *Parabacteroides* genus and Aß_42_ only in MCI participants on the MKD diet, while the genus *Oscillospira* was positively correlated with Aß_42_ in CN participants on the MKD diet.

The MKD diet may therefore modulate the gut microbiome and bacterial metabolites and thus influence AD biomarkers in cerebrospinal fluid. Therefore, the combination of an antioxidant Mediterranean diet with a state of ketosis and carbohydrate restriction appears to be a very good alternative to the standard classic ketogenic diet, which is dominated by saturated fats [122].

Summary:



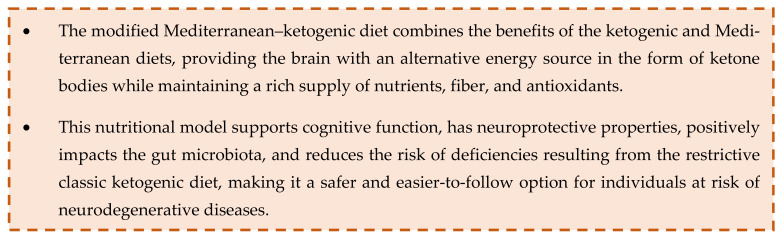



### 4.6. The Western-Style Diet (The Western Diet)

The Western diet (WD) is characterized by the consumption of foods that are low in nutritional value but high in calories. The WD is rich in fat, including saturated fatty acids, simple sugars, and salt and low in fiber. A typical American diet derives 35% of its energy from fat, 49% from carbohydrates, and 16% from protein [110]. A typical WD consists of processed fast food, red meat, sweet desserts, high-fat foods, full-fat dairy products (whole milk, butter, cream, full-fat cheeses, cottage cheese, yogurt, and kefir), sugar-rich drinks, and excessive amounts of fried foods (Table 1). High consumption of saturated fats, including excessive amounts of omega-6 polyunsaturated fatty acids and low amounts of omega-3, and an unhealthy omega-6/omega-3 ratio of 20:1 are detrimental to health. In addition, AGEs found in heat-treated foods (frying, baking, grilling) have a negative effect on the proper functioning of the body. Following this diet can lead to an increased incidence of many diseases, primarily obesity, type 2 diabetes, dyslipidemia, inflammatory bowel disease, certain types of cancer, cardiovascular disease, excessive stimulation of the renin–angiotensin system, and inflammation in the nervous system that may contribute to the development of neurodegenerative disease. In addition, such a diet negatively affects the diversity and activity of the intestinal bacterial flora, leading to dysbiosis and intestinal barrier dysfunction and increased permeability and leakage of toxic bacterial metabolites into the circulation and brain [16]. Another consequence of WD and accompanying dysbiosis is impaired production by intestinal bacteria of neurotransmitters, such as GABA, tryptophan, serotonin, and dopamine [124]. People who follow a WD, which is typically low in vegetables, fruit, whole grains, and other plant products, are at risk of fiber deficiency, which is very important for the composition of the intestinal microflora. The most numerous bacteria in the large intestine are protective bacteria, such as *Bifidobacterium* ssp., *Bacterioides* ssp., and *Lactobacillus* spp. In contrast, a WD causes significant changes in the intestinal microflora, namely an increase in the number of species from the *Enterobacteriaceae* family (*Escherichia*, *Shigella*) and a decrease in bacteria of the genus *Bifidobacterium*, *Lactobacillus*, *Eubacterium*, and *Bacteroides* [125,126].

The Western diet is associated with increased levels of trimethylamine N-oxide (TMAO). TMAO is a pro-inflammatory toxin derived from the intestinal metabolism of choline, betaine, and carnitine, the main sources of which are meat, fish, eggs, and dairy products. These components are metabolized in the large intestine mainly by bacteria belonging to the Firmicutes (Bacillota) and Proteobacteria phyla to produce trimethylamine (TMA), which is then oxidized to TMAO in the liver by flavin monooxygenases 3 (FMO3). Studies indicate that TMAO may contribute to accelerated neurodegeneration and Alzheimer’s disease. Higher concentrations of TMAO have been observed in the cerebrospinal fluid of patients with MCI and in people with AD compared to healthy individuals with normal cognitive function [127].

In individuals with high intake of animal products and low fiber intake, there is an increase in the number of bacteria producing proteases of the genus *Alistipes*, *Bilophila*, and *Bacteroides* and a decrease in the number of Firmicutes (Bacillota) involved in the breakdown of plant polysaccharides, producing butyrate and propionate, i.e., *Roseburia*, *Eubacterium rectale* and *Ruminococcus bromii* [16,124,128,129].

The WD, high in fat and simple sugars, is also associated with a decrease in the number of lactic-acid-producing bacteria, i.e., *Lactobacillus* and *Streptococcus*, as well as a decrease in bacteria belonging to the *Prevotellaceae* family (involved in the production of mucin in the intestinal mucosa and short-chain fatty acids) and the *Rikenellaceae* family (involved in the synthesis of propionate, acetate, and/or succinate) [125]. The WD also causes an increase in bacteria of the genus *Desulfovibrio*, *Desulfobacter*, *Desulfomonas*, *Desulfobulbus*, and *Desulfotomaculum*, which reduce sulphates, and excess hydrogen sulfide production disrupts the normal function of the intestinal barrier, increasing its permeability [130]. In addition, the WD promotes the growth of the Proteobacteria phylum, and bacteria belonging to the *Enterobacteriaceae* family are pathogenic bacteria. Their overproduction of LPS is an important pathological factor (infections caused by them are characterized by a severe course) [16,131].

Changes in gut bacteria species are described using the Firmicutes (Bacillota)/Bacteroidetes ratio, which is a marker of microbiome dynamics. Studies in animal and human models have shown that a high-fat diet increases the Firmicutes (Bacillota)/Bacteroidetes ratio due to an increase in the abundance of, among other classes, *Erysipelotrichia*, *Bacilli*, *Clostridia*, and the genera belonging to it, e.g., *Dorea and Ruminococcus.* There is also an increase in the abundance of Gram-negative *Proteobacteria*, which are bacteria that produce the most potent pro-inflammatory endotoxin, LPS [16,132]. In an experimental study in a mouse model, it was observed that on a high-fat diet, the intestinal microflora of mice had reduced activity of metabolic processes, including nucleotide biosynthesis, glycolysis, gluconeogenesis, starch degradation, and pyruvate fermentation, compared to control mice fed a low-fat diet [126]. Other authors, Jena et al. [132], in an experiment on mice, showed that a WD induced dysbiosis and dysregulation of BA synthesis with a reduction in endogenous ligands for BA receptors in the liver and brain, i.e., farnesoid X receptor (FXR) (responsible for the balance of many neurotransmitters in various areas of the brain and a neurobehavioral regulator) and G-coupled protein receptor specific for bile acids (TGR5), which controls the production of peptide YY (PYY) and glucagon-like peptide 1 (GLP1), thereby regulating appropriate food intake and insulin sensitivity. Dysregulated BA synthesis and dysbiosis were accompanied by systemic inflammation, as well as microgliosis and reduced neuroplasticity. A reduced abundance of bacteria of the phylum Bacteroidetes (decrease in the families *Bacteroidaceae* and *Porphyromonadaceae*) and a reduction in the number of bacteria of the phylum Actinobacteria (decrease in the families *Bifidobacteriaceae* and *Coriobacteriaceae*) were also found and an abundance of the phylum Firmicutes (Bacillota) (increase in the *Lachnospiraceae*, *Clostridiaceae*, and *Peptostreptococcaceae* families). An increase in *Desulfovibrionaceae* (phylum Thermodesulfobacteriota) and *Deferribacteraceae* (phylum Deferribacterota) was also observed. Another study conducted by Graham et al. [133] on a mouse model showed a negative effect of the WD on brain function. In rodents fed a WD for 8 months, increased neuroinflammatory status and increased neuronal cell loss were observed, as well as increased Aβ deposition and increased activation of microglia and monocytes in response to Aβ accumulation. The authors of the study noted that a long-term WD led to a significant increase in the immune response in the brains of all mice, which, according to the authors, may increase susceptibility to AD. A unique achievement of the study was establishing a relationship between the increase in the number of microglia and monocytes expressing triggering receptor expressed on myeloid cells 2 (TREM2) on their surface and the increased amount of amyloid plaques in the brains of the studied mice. It is known that the gene encoding the TREM2 protein, as an Aβ microglial receptor, mediates both physiological and pathological effects associated with Aβ- in Alzheimer’s disease (excessive microglia activation accelerates Aβ accumulation in the late stages of Alzheimer’s disease and other age-related neurodegenerative diseases). The interaction between microglia and Aβ accumulation leads to a vicious cycle; instead of effectively removing Aβ deposits, microglia contribute to neuroinflammation and cell damage. Understanding this interaction is crucial, as researchers are seeking ways to modulate microglial activity to clear Aβ deposits and inhibit neuroinflammation, which could slow disease progression. The study by Prakash P, et al. [134] showed that microglia formed lipid droplets (LD) after exposure to Aβ and that the amount of LD increased in the brains of people with AD. Microglia loaded with LD showed defects in Aβ phagocytosis, and a decrease in free fatty acids (FFA) and an increase in triacylglycerols (TG), compounds involved in LD formation, were also demonstrated. It turns out that diacylglycerol O-acyltransferase 2 (DGAT2)—which catalyzes the final stage of TG synthesis, a key enzyme that converts FFA to TG—was increased in the brains of people with AD. The study indicates that DGAT2 inhibition is a promising target for the treatment of Alzheimer’s disease compared to other LD-reducing compounds [134].

One study showed that a WD in adults with MCI may reduce the risk of developing AD. In a study conducted by Hoscheidt et al. [89], an increase in the Aβ_42_/_40_ ratio and decreased t-tau protein concentration and Aβ_42_/t-tau ratio were observed, which may indicate a lower risk of AD. The results of the study should be supported by more evidence.

Summary:



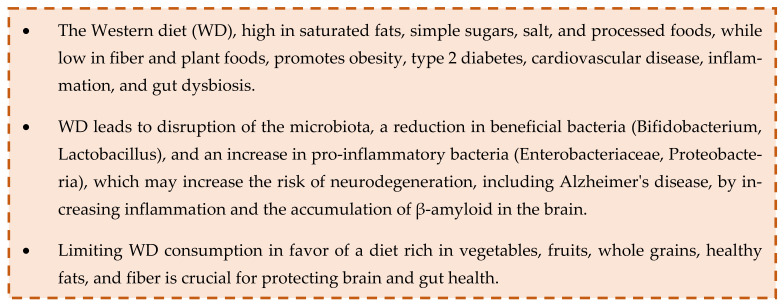



## 5. Conclusions

Studies on the relationship between diet and gut microbiota suggest that diet and the gut microbiome may play a role in the prevention of neurodegenerative diseases, including AD. The composition of the gut microbiota is influenced not only by dietary patterns, individual predispositions and lifestyle, medications and stimulants, but also by coexisting diseases and many other factors. A growing body of research indicates that diet-dependent changes in the composition of the gut microbiota may adversely affect brain physiology and increase the risk of AD. An imbalance in the gut microbiota ecosystem, which protects bacteria that secrete large amounts of amyloid and lipopolysaccharide, may contribute to the activation of signaling pathways and the production of pro-inflammatory cytokines, which play a role in neurodegeneration and the pathogenesis of AD. The contribution of the gut microbiota to amyloid formation becomes even greater with aging, due to the increased permeability of the intestinal epithelium and the blood-brain barrier to small molecules, including amyloid. In addition, the composition of the gut microbiota changes during aging. These changes may lead to increased inflammation of the nervous system and dysfunction of specific areas of the brain, such as the cerebellum and hippocampus, which are characteristic of Alzheimer’s disease.

A healthy diet that is good for the diversity of the gut ecosystem, based on natural plant products with probiotics, prebiotics, and other compounds, and limiting red meat, processed foods, simple sugars, and saturated fatty acids and avoids trans fatty acids, plays an important role in the prevention of neurodegenerative processes in AD. Although there are no ideal food combinations for a healthy diet for the microbiome and brain, several dietary models may have significant preventive and therapeutic effects in AD. A dietary model based on the principles of the MedDiet is believed to prevent the development of AD. In addition, the DASH and MIND diets protect the brain thanks to their high content of neuroprotective bioactive compounds found in green leafy vegetables, fruits, nuts, seeds and legumes, which protect neurons from damage associated with oxidative stress caused by free radicals. Many studies indicate that the KD improves brain function, has a neuroprotective effect on aging brain cells, improves circulation in the blood vessels of the brain, leading to a significant improvement in cognitive function due to increased uptake of ketone bodies into the brain. However, following this diet leads to a significant decrease in bacterial diversity and quantitative disturbances of health-promoting bacteria in favor of harmful bacteria. Long-term use of the KD may consequently lead to increased intestinal barrier permeability and the penetration of harmful microorganisms and their metabolites, which have a dangerous effect on the brain. An alternative to the KD is a dietary model based on the principles of the MedDiet and KD, known as the MKD, which increases the diversity of gut bacteria and thus improves brain function. WD has an adverse effect on the gut microbiota, altering the composition and abundance of the gut microbiome, leading to many diseases, including neurodegenerative diseases.

Based on the results of numerous studies, it can be concluded that a diet containing specific nutrients and rich in prebiotics and probiotics is an undisputed factor that can modulate the gut microbiome and thus prevent, delay, and alleviate the symptoms of AD. Although the role of the gut microbiota in the development of AD still requires further research, it is believed that a balanced diet will maintain eubiosis and the proper functioning of the gut–brain axis in a way that is beneficial to brain health.

Promoting sustainable diets is key to counteracting the negative consequences of the global food system. The healthcare system, dietary counseling, and social programs can be effective tools for educating the public by guiding their diets toward sustainable development. Healthcare professionals, especially dietitians, who can advise patients on dietary modifications, develop nutrition plans, and provide dietary advice, should play an important role in promoting a healthy lifestyle. Personalized nutrition programs should play an important role in the healthcare system due to the increasing prevalence of neurodegenerative diseases associated with our society’s lifestyle. In the future, dietary counseling will be based on recommendations prepared for population groups with common genetic characteristics, which will enable the development of nutritional advice and health prevention programs tailored to the microbiome of a specific person or group of people.

## Figures and Tables

**Figure 1 nutrients-17-03053-f001:**
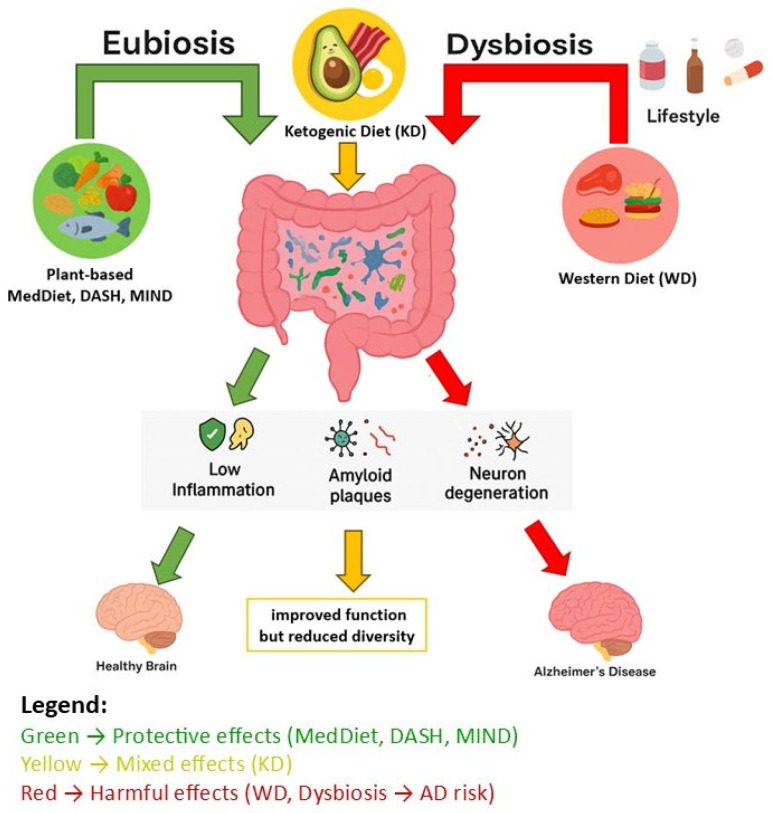
Diet, gut microbiota, and Alzheimer’s disease risk. Interactions and preventive potential.

**Table 1 nutrients-17-03053-t001:** Dietary components affecting the gut microbiota and cognitive function and dementia risk.

Dietary Components	High Amounts	Moderate Amounts	Small Amounts (Restricted)
Fish (unfried)	×	×	
Extra- virgin olive oil (cold-pressed)	×		
Fresh vegetables, including green leafy vegetables	×		
Fresh fruit, including berries	×		
Wholegrain cereal products	×		
Fermented milk products (kefir, yogurt), sauerkraut and cucumbers, kimchi, natto, miso	×		
Low-fat dairy products	×		
Nuts	×		
Seeds (pumpkin seeds, sunflower seeds)	×		
Water	×		
Herbs and spices (especially those with anti-inflammatory properties)	×		
Legumes	×		
Dairy products		×	
Poultry (unfried)		×	
Eggs		×	
Alcohol (particularly red wine, with meals)		×	×
Potatoes and sweet potatoes		×	
Other liquids (tea, coffee unsweetened), natural fruit juices		×	
Red meat, processed meat products, and offal			×
Fatty dairy			×
Butter, margarine			×
Cheese			×
Fried foods			×
Sweets and sweet snacks, confectioneries			×
Highly processed food			×
Fast food			×
Salt			×
Sweetened drinks			×

×, indicates which products are recommended in high and moderate quantities, and which should be limited.

## Data Availability

No new data were created or analyzed in this study. Data sharing is not applicable to this article.

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
