# Peer review of "Diet as a Modulator of Gut Microbiota May Reduce Alzheimer’s Disease Risk"

_nutrients, 2025, doi:10.3390/nu17193053_

Round 1
Reviewer 1 Report
Comments and Suggestions for Authors
The manuscript provides an impressive and comprehensive overview of dietary models affecting Alzheimer’s disease and cognitive function, effectively integrating human, animal, and cell-based studies The discussion on the gut microbiome is up-to-date, and Table 1 clearly organizes recommended and restricted foods. However, clinical relevance, population differences (age, sex, genetic risk factors such as APOE4, cultural dietary habits), and long-term dietary effects are insufficiently addressed. The text is technically accurate but dense, with few brief summaries or visual elements. The methodology section is missing, leaving study selection and quality assessment unclear.
Recommendations for the authors:
- Add a 2–3 sentence summary and practical recommendation at the end of each diet section.
- Make the text more accessible with figures, mechanism diagrams / tables, and highlighted key points.
- Include a brief methodology section: search strategy, databases, keywords, inclusion/exclusion criteria, and quality assessment of sources.
- Highlight clinical relevance and population differences, including genetic risk factors (e.g APOE4 carriers), and cultural and lifestyle considerations affecting diet adherence.
- Discuss potential long-term effects, such as micronutrient deficiencies, prolonged ketone or MCT diet impacts (metabolism, kidney function, bone health), and other possible side effects.
- Address practical applicability: ease of following diets in real-life settings, strategies to improve adherence (dietary counseling, community programs), and the role of dietitians and healthcare professionals in safe, personalized implementation.
Author Response
Thank you very much for your valuable comments on this manuscript.
The gut microbiota and diet have a huge impact on the prevention (delay) of many diseases, including Alzheimer's disease, which is the subject of this publication. The publication submitted for review shows that diet and the gut microbiome are a closed circle: what we eat affects the development of the gut microbiome, and the microbiome affects our overall health and, above all, the health of our brain.
Thank you to the reviewer for raising a very interesting point about cultural eating habits. Established eating habits, shaped by tradition, the environment, and social norms in which we live (including family) and local eating habits, which include not only what we eat, but also how we prepare and consume food, which is related to food culture, have a huge impact on our diet.
Below, answers to your comments, I hope you find them fully satisfactory.
Comments 1: Add a 2–3 sentence summary and practical recommendation at the end of each diet section.
Response: Each chapter has been summarized.
Chapter 2: Disorders in the development of the gut microbiome affect not only the functioning and development of the digestive system, and immune systems but also the nervous. The cross-talk between gut microbiota and brain may have crucial impact in neurodegenerative disorders.
Chapter 3: As a result of dysbiosis and the growth of pro-inflammatory strains, and additionally, the action of bacterial amyloid, inflammation develops in the CNS, which predisposes to the faster onset of AD symptoms. In older people, the role of the gut microbiota in amyloid formation becomes more significant because small molecules easily penetrate the more permeable intestinal epithelium and the blood-brain barier.
Chapter 4: According to many studies, the best dietary pattern for brain function is the Mediterranean diet, a modified ketogenic-Mediterranean diet, DASH and MIND, which may prevent and slow the development of AD.
Additionally, a short summary has been prepared for each diet in the form of bullet points in a box:
4.1. The Mediterranean diet
4.2. The Dietary Approaches to Stop Hypertension diet
4.3. The Mediterranean-DASH Intervention for Neurodegenerative Delay diet
4.4. The Ketogenic Diet
4.5. The Modified Mediterranean-ketogenic diet
4.6. The Western-Style Diet (The Western Diet)
Comments 2: Make the text more accessible with figures, mechanism diagrams / tables, and highlighted key points.
Response: We added Figure 1.
Comments 3: Include a brief methodology section: search strategy, databases, keywords, inclusion/exclusion criteria, and quality assessment of sources.
Response: We added at the end of the Chapter 1: The methods of this review article were based on the use of the PubMed, Cochrane, EBSCO, EMBASE and Scopus database, to search for all related published studies. The selection was based on the keywords, “ Alzheimer’s disease”,“ diet”, “ gut-brain axis”, “ microbiome”, “ neurodegeneration”.
Comments 4: Highlight clinical relevance and population differences, including genetic risk factors (e.g APOE4 carriers), and cultural and lifestyle considerations affecting diet adherence.
Response: In the publication, the authors concentrated exclusively on people with AD, which most commonly affects people over the age of 65, and its risk increases with age. Alzheimer's disease is not characterized by distinct symptoms based on gender, as the symptoms are the same in women and men, but women are more likely to develop the disease and experience a slightly more severe course, which may be related to longer life expectancy, hormonal changes after menopause, and genetic and social factors. However, these are not differences in the symptoms themselves, but in the frequency of occurrence and intensity of symptoms. In the introduction, the authors mentioned the etiopathogenesis of the disease, which is multifactorial in nature, including a genetic defect (approx. 5% of AD cases), such as the APOE 4 defect, which increases the risk of developing the disease by up to several times.
Comments 5: Discuss potential long-term effects, such as micronutrient deficiencies, prolonged ketone or MCT diet impacts (metabolism, kidney function, bone health), and other possible side effects.
Response: The authors describe the side effects of the ketogenic diet in section 4.4.: Following the KD diet is not easy and requires constant medical supervision. Restricting the consumption of carbohydrates contained in fibre-rich foods (e.g. legumes, whole grains), which are essential for the intestinal bacterial ecosystem, can be harmful to overall health. Therefore, to ensure that the diet is followed correctly, ketone body concentrations in urine should be monitored. In addition, it is recommended to check glucose, albumin, total protein, total cholesterol, triglycerides and creatinine levels once every 3 months. Once a year, an ultrasound examination of the kidneys, bone density, carnitine, selenium and an electrocardiogram should be performed, which are important in preventing long-term effects such as kidney stones, osteoporosis, hyperlipidaemia, carnitine deficiency and cardiomyopathy.
The authors describe the side effects of the Western diet in section 4.6.: Following this diet can lead to an increased incidence of many diseases, primarily obesity, type 2 diabetes, dyslipidaemia, inflammatory bowel disease, certain types of cancer, cardiovascular disease, excessive stimulation of the renin-angiotensin system, and inflammation in the nervous system that may contribute to the development of neurodegenerative disease. In addition, such a diet negatively affects the diversity and activity of the intestinal bacterial flora, leading to dysbiosis and intestinal barrier dysfunction, increased permeability and leakage of toxic bacterial metabolites into the circulation and brain.
What are the long-term effects of the diet? The long-term effects of the diet can be negative, including nutritional deficiencies, metabolic problems, hormonal disorders, mental health deterioration (depression, anxiety, eating disorders), and an increased risk of developing diseases such as type 2 diabetes and heart disease. Overly restrictive “miracle” diets can lead to hunger, loss of muscle mass, concentration problems (brain fog), and social isolation, so a balanced diet tailored to individual needs is crucial. The Western-style diet is not the preferred nutritional model for any age group. According to many studies, the best dietary pattern that has a positive impact on general health, including brain function, are the Mediterranean diet, the modified ketogenic-Mediterranean diet, DASH, and MIND, which may prevent and slow down the development of AD. These diets and their impact on the role and development of gut bacteria are discussed in detail in the individual subsections.
Comments 6: Address practical applicability: ease of following diets in real-life settings, strategies to improve adherence (dietary counseling, community programs), and the role of dietitians and healthcare professionals in safe, personalized implementation.
Response: At the end of chapter 5, we added: Promoting sustainable diets is key to counteracting the negative consequences of the global food system. The healthcare system, dietary counseling, and social programs can be effective tools for educating the public by guiding their diets toward sustainable development. Healthcare professionals, especially dietitians, who can advise patients on dietary modifications, develop nutrition plans, and provide dietary advice, should play an important role in promoting a healthy lifestyle. Personalized nutrition programs should play an important role in a healthcare system due to the increasing prevalence of neurodegenerative diseases associated with our society's lifestyle.
The authors are currently preparing material for a chapter in a book on the topics suggested by the reviewer. The reviewer's suggestions for expanding knowledge mentioned above will be discussed in detail. This manuscript focuses mainly on one disease entity, Alzheimer's disease, and the role of gut bacteria in various dietary models and their impact on brain health in AD patients.
Reviewer 2 Report
Comments and Suggestions for Authors
I congratulate the Authors for this huge work
The paper covers many topics in depth, and it's enjoyable to read
I have some comments that I hope can improve the quality of the paper
Generally speaking, as I have repeatedly reported, the good role of alcohol must be resized. Accordingly to recent evidences, alcohol should not be recommended in any case, and I suggest that this trend must be followed in the review. Furthermore, the quality of the cited literature should be increased, with more reviews, meta-analysis, systematic reviews and not mainly singular studies or animal research.
line 86 You can add mitochondrial causes between causes of AD, i.e. one of the reason why KD works
line 131 Studies show alcohol has a detrimental effect on the brain, even in small quantities. Please exclude wine and alcohol from healthy foods.
line 175 pay attention to the words
line 392 you can add a new research that involve microglia and lipid accumulation in the genesis of Alzheimer disease. ref https://www.cell.com/immunity/abstract/S1074-7613(25)00192-X
line 408 pay attention to the repeated words
line 421 and following please rephrase this paragraph, I think there may be typos
line 537 missing letters?
line 558 are you listing some diets? The phrase must be corrected
line 616 It's true that red wine is associated with better microbiota, but we still should not recommend alcohol consumption, if not occasionally, according to more recent evidence. Please underline this aspect; ref https://pubmed.ncbi.nlm.nih.gov/40087038/
line 679 please specify better that DASH diet is similar to MedDiet
line 722 "alcohol is good for brain" is only a hypothesis, disproved by recent evidence. This trend in the review must be corrected https://www.nature.com/articles/s41467-022-28735-5
line 773 once again, wine consume is an individual option, even in MIND diet, and should not be promoted
line 780 and 787 It's better to talk about "folates" and not "folic acid", an artificial molecule
line 816 it appears a bit redundant. You can write: as for Med and DASH Diet, prebiotics are important source of healthy nutrients...
line 840 please specify that the crossing of BBB happens when [KB] is sufficiently high
line 862 please add "in small quantity" with respect to fruit
line 870 in urine or in blood
line 1020 please reformulate the phrase with the acronym (AHAD) after American Heart Association diet
line 1055 dairy are not necessarily full-fat
line 1057 you can add excess of AGEs as feature of WD, you cited them in the first part of paper
line 1198 please add "and avoid trans fatty acids"
line 1229 You can add a note about the customization of the diet; the best diet for one individual is not necessarily for another one, in a way to respect personal tastes and outcomes, that can be dependent by interactions between original microbiota and individual genetic
A graphical abstract should improve the attractiveness of the paper
Author Response
Thank you very much for your valuable comments on this manuscript.
Availability of publications in the largest database, PubMed (and many others), after writing the keywords: microbiota, diet, Alzheimer's disease, meta-analysis, 23 records were obtained, including studies conducted on animal models and cell lines. Writing the words “systematic reviews” yielded 10 records. There are few publications on human studies that have examined the relationship between types of diet, the gut ecosystem, and Alzheimer's disease. Therefore, these few publications have been cited in this publication.
The topic of alcohol consumption, even in moderate amounts (a glass of wine with dinner), is also problematic. The Mediterranean diet allows moderate alcohol consumption, most often in the form of red wine with a meal, but this is not mandatory and heavy drinking or excessive amounts are not recommended. The DASH diet recommends eliminating alcohol. The MIND diet allows the consumption of one glass of wine a day as one of ten recommended products that have a positive effect on the cognitive functions of the brain. Research by other authors suggests that low or moderate alcohol consumption is associated with a lower risk of dementia and Alzheimer's disease [ Porras-García E, Fernández-Espada Calderón I, Gavala-González J, Fernández-García JC. Potential neuroprotective effects of fermented foods and beverages in old age: a systematic review. Front Nutr. 2023;10:1170841. doi: 10.3389/fnut.2023.1170841]. In very small amounts, alcohol may have certain health benefits, such as improving the lipid profile (lowering “bad” cholesterol, increasing “good” cholesterol) and reducing the risk of cardiovascular disease and stroke. The polyphenols (especially resveratrol) found in some alcoholic beverages, such as red wine, act as antioxidants. According to another group of authors, even the smallest amount of alcohol is not recommended, as it increases the risk of diseases, especially cancer and heart disease, and damages the brain. The concept of a “safe” dose of alcohol is increasingly being questioned, and studies indicate that any amount of ethanol can be harmful. Alcohol negatively affects the nervous system, liver, and brain, and its consumption is associated with a higher risk of systemic diseases.
Below, answers to your comments, I hope you find them fully satisfactory.
Answers:
Line 86. You can add mitochondrial causes between causes of AD, i.e. one of the reason why KD works
We added: In addition to the factors mentioned above, mitochondrial dysfunction is also a mediator of Alzheimer's disease pathogenesis. Mitochondria, cellular organelles, are responsible for producing the energy necessary for neurons to function. Mitochondrial dysfunction involves abnormal structure and dynamics, as well as disturbances in the functioning of the electron transport chain, the formation of excessive amounts of free radicals, and oxidative stress. Impaired levels of enzymes involved in the electron transport chain and the Krebs cycle can escalate pathological processes. Mitochondria play a key role in regulating calcium balance and apoptosis. Their dysfunction can lead to the development of inflammatory processes and progressive damage to nerve tissue.
Line 131. Studies show alcohol has a detrimental effect on the brain, even in small quantities. Please exclude wine and alcohol from healthy foods.
We excluded wine and alcohol from healthy foods
Line 175 pay attention to the words
We removed nd
Line 392 you can add a new research that involve microglia and lipid accumulation in the genesis of Alzheimer disease. ref https://www.cell.com/immunity/abstract/S1074-7613(25)00192-X
We have cited this publication at the end of section 4.6, no. 134
Line 408 pay attention to the repeated words
We changed significant to serious
Line 421 and following please rephrase this paragraph, I think there may be typos
We added Enterococcus spp.
Line 537 missing letters?
The sentence was corrected
Line 558 are you listing some diets? The phrase must be corrected
We added: Studies have shown that a Western diet ( characterized by high consumption of saturated fats, sugar, salt, and low consumption of fiber, vegetables and fruit) causes a significant reduction in microorganisms of the genus Bifidobacterium, Bacteroides, Prevotella and bacteria producing butyric acid, mainly of the genus Clostridium, Eubacterium and Fusobacterium, with a simultaneous increase in Firmicutes (Bacillota).
Line 616 It's true that red wine is associated with better microbiota, but we still should not recommend alcohol consumption, if not occasionally, according to more recent evidence. Please underline this aspect; ref https://pubmed.ncbi.nlm.nih.gov/40087038/
We added: Alcohol consumption is neither a necessary nor a recommended part of the diet. It is believed that there is no safe dose of alcohol. Even small amounts can have a negative impact on health.
Line 679 please specify better that DASH diet is similar to MedDiet
The similarities between the two diets (MedDiet and DASH) are listed in section 4.2, lines 684-694.
Line 722 "alcohol is good for brain" is only a hypothesis, disproved by recent evidence. This trend in the review must be corrected https://www.nature.com/articles/s41467-022-28735-5
And
line 773 once again, wine consume is an individual option, even in MIND diet, and should not be promoted:
We are not advocates of drinking alcohol. We believe that even the smallest amounts of alcohol are harmful to health. We cite studies conducted by other authors.
Line 780 and 787 It's better to talk about "folates" and not "folic acid", an artificial molecule :
was changed to: folates (natural form of vitamin B9)
Line 816 it appears a bit redundant. You can write: as for Med and DASH Diet, prebiotics are important source of healthy nutrients
However, the authors decided to save the listed sources of prebiotics. Readers will not be forced to look for other sources of information.
Line 840 please specify that the crossing of BBB happens when [KB] is sufficiently high
We added; A significant increase in the concentration of...
Line 862 please add "in small quantity" with respect to fruit
We added: in small quantity
Line 870 in urine or in blood
We added as you suggested.
Line 1020 please reformulate the phrase with the acronym (AHAD) after American Heart Association diet
We added: The target macronutrient composition (% of total calories) for the AHAD was 55–65% carbohydrate, 15–20% fat, and 20–30% protein (for the MKD was <10% carbohydrate, 60–65% fat, and 30–35% protein).
Line 1055 dairy are not necessarily full-fat
We added: (whole milk, butter, cream, full-fat cheeses, cottage cheese, yogurt, and kefir).
Line 1057 you can add excess of AGEs as feature of WD, you cited them in the first part of paper We added: In addition, AGEs found in heat-treated foods (frying, baking, grilling) have a negative effect on the proper functioning of the body.
Line 1198 please add "and avoid trans fatty acids"
We added: and avoid trans fatty acids
Line 1229 You can add a note about the customization of the diet; the best diet for one individual is not necessarily for another one, in a way to respect personal tastes and outcomes, that can be dependent by interactions between original microbiota and individual genetic
We added: Personalized nutrition programs should play an important role in a healthcare system due to the increasing prevalence of neurodegenerative diseases associated with our society's lifestyle. In the future, dietary counseling will be based on recommendations prepared for population groups with common genetic characteristics, which will enable the development of nutritional advice and health prevention programs tailored to the microbiome of a specific person or group of people.
A graphical abstract should improve the attractiveness of the paper
We added Figure 1
Thank you once again for your insightful review of our article.
Round 2
Reviewer 2 Report
Comments and Suggestions for Authors
The quality of the work has significantly improved.
the added figure is very nice.
Authors took my suggestions into account and properly correct the paper.
I have just very few points to note
line 295 please correct the name of molecules, CO2, H2 and H2S, with right spaces and layout
line 574 the sentence is unclear, maybe any word missing?
line 880 missing "F" in fruits
Once again congratulations for this work